# The *PINK1* p.Asn521Thr Variant Is Associated with Earlier Disease Onset in *GRN/C9orf72* Frontotemporal Lobar Degeneration

**DOI:** 10.3390/ijms232112847

**Published:** 2022-10-25

**Authors:** Giacomina Rossi, Erika Salvi, Luisa Benussi, Elkadia Mehmeti, Andrea Geviti, Sonia Bellini, Antonio Longobardi, Alessandro Facconi, Matteo Carrara, Cristian Bonvicini, Roland Nicsanu, Claudia Saraceno, Martina Ricci, Giorgio Giaccone, Giuliano Binetti, Roberta Ghidoni

**Affiliations:** 1Neurology V and Neuropathology Unit, Fondazione IRCCS Istituto Neurologico Carlo Besta, 20133 Milan, Italy; 2Neuroalgology Unit, Fondazione IRCCS Istituto Neurologico Carlo Besta, 20133 Milan, Italy; 3Molecular Markers Laboratory, IRCCS Istituto Centro San Giovanni di Dio Fatebenefratelli, 25125 Brescia, Italy; 4Service of Statistics, IRCCS Istituto Centro San Giovanni di Dio Fatebenefratelli, 25125 Brescia, Italy; 5MAC-Memory Clinic and Molecular Markers Laboratory, IRCCS Istituto Centro San Giovanni di Dio Fatebenefratelli, 25125 Brescia, Italy

**Keywords:** genetic frontotemporal lobar degeneration, *GRN*, *C9orf72*, age of onset, *PINK1*, disease modulators

## Abstract

Genetic frontotemporal lobar degeneration (FTLD) is characterized by heterogeneous phenotypic expression, with a disease onset highly variable even in patients carrying the same mutation. Herein we investigated if variants in lysosomal genes modulate the age of onset both in FTLD due to *GRN* null mutations and *C9orf72* expansion. In a total of 127 subjects (*n* = 74 *GRN* mutations and *n* = 53 *C9orf72* expansion carriers), we performed targeted sequencing of the top 98 genes belonging to the lysosomal pathway, selected based on their high expression in multiple brain regions. We described an earlier disease onset in *GRN/C9orf72* pedigrees in subjects carrying the p.Asn521Thr variant (rs1043424) in PTEN-induced kinase 1 (*PINK1*), a gene that is already known to be involved in neurodegenerative diseases. We found that: (i) the *PINK1* rs1043424 C allele is significantly associated with the age of onset; (ii) every risk C allele increases hazard by 2.11%; (iii) the estimated median age of onset in homozygous risk allele carriers is 10–12 years earlier than heterozygous/wild type homozygous subjects. A replication study in *GRN/C9orf72* negative FTLD patients confirmed that the rs1043424 C allele was associated with earlier disease onset (−5.5 years in CC versus A carriers). Understanding the potential mechanisms behind the observed modulating effect of the *PINK1* gene in FTLD might prove critical for identifying biomarkers and/or designing drugs to modify the age of onset, especially in *GRN/C9orf72*-driven disease.

## 1. Introduction

Frontotemporal lobar degeneration (FTLD) represents one of the most common forms of early-onset dementia while accounting for up to 15% of all dementias [1,2]. FTLD is neuropathologically characterized by the accumulation of different proteins: microtubule-associated protein tau (MAPT), TAR DNA-binding protein 43 (TDP-43), and the proteins belonging to the FET group, consisting of fused in sarcoma (FUS), Ewing’s sarcoma protein (EWS), and TATA-binding protein-associated factor 2N (TAF15). Clinically, FTLD patients show degeneration in the frontal and temporal lobes resulting in progressive behavior, personality, executive function, and language deficits [3,4]. While sporadic forms of FTLD without a clear genetic etiology are still poorly understood, 30–50% of FTLD patients present a positive family history of dementia, and most of them are carriers of mutations in genes associated with the pathology [5,6,7].

Regarding the FTLD cases not explained by a single pathogenic mutation, previous GWAS studies reported that multiple genetic risk loci are associated with the disorder [8,9], thus suggesting FTLD as a polygenic disease.

Regarding the monogenic forms, the most common causative genes in FTLD are the chromosome 9 open reading frame *(C9orf72*) [10,11], progranulin *(GRN*) [12,13], and tau (*MAPT*) [14,15]. The most common cause of familial FTLD is represented by pathological expansions (>30) of a hexanucleotide repeat (GGGGCC) in the first intron/promoter of the *C9orf72* gene [10,11]; intermediate expansions (12–30 hexanucleotide repeats) have a risk effect in familial/sporadic FTLD and could influence the age of onset and clinical subtype [16,17]. C9orf72 protein is thought to play a role in autophagy and endosomal trafficking. Three disease mechanisms caused by hexanucleotide expansions have been proposed: (i) haploinsufficiency due to reduced expression of C9orf72 from the expanded allele, (ii) formation of toxic repeat-containing RNA foci, and (iii) production of toxic dipeptide repeat proteins aggregates. These mechanisms can lead to a variety of cell dysfunctions, including endolysosomal and/or autophagic defects [18].

*GRN* mutations account for up to 25% of familial FTLD cases [19], with most of the pathogenic mutations being *null* ones introducing premature termination codons and leading to a reduction of circulating progranulin [20]. Progranulin is a growth factor involved in inflammation, wound healing, and cancer; in the central nervous system, it modulates inflammation and acts as a neurotrophic and neuroprotective factor [21]. Increasing evidence indicates that progranulin has a role in lysosomes, where it activates cathepsin D and also affects glucocerebrosidase activity [21]. Rare homozygous *GRN* mutations mostly cause neuronal ceroid lipofuscinosis (NCL), a lysosomal storage disorder that shares some neuropathological features with FTLD caused by heterozygous *GRN* mutations [22,23]. Thus, FTLD due to *GRN* null mutations or *C9orf72* expansions seems to share both molecular mechanisms, such as reduction of the functional proteins [10,11,12,13,18,20], and pathological conditions, such as inflammation caused by microglia activation and lysosomal dysfunction [24,25].

Genetic FTLD is characterized by heterogeneous phenotypic expression, with disease onset, age of death, and disease duration highly variable even in patients carrying the same mutations, in particular in *GRN/C9orf72* pedigrees [26]. Recent GWAS studies identified potential genetic modifiers of disease risk and age of onset in FTLD patients with *GRN* mutations [27] or *C9orf72* expansions [28,29]. In addition, some reports suggested the possibility that this phenotypic variability may have a polygenic basis, due to actual findings of FTD cases with a digenic disease, as carrying a pathogenic mutation in *GRN* or *MAPT* together with *C9orf72* expansion [30,31,32].

Herein, we investigated if variants in lysosomal genes modulate age of onset in FTLD due to *GRN* null mutations or *C9orf72* expansions. To reach this goal, we performed, in a large group of *GRN* null mutations and *C9orf72* expansion carriers, targeted sequencing of the top 98 genes belonging to the lysosomal pathway, selected based on their high expression in multiple brain regions. We described an earlier disease onset in *GRN/C9orf72* pedigrees in subjects carrying the p.Asn521Thr variant in PTEN-induced kinase 1 (*PINK1*), thus highlighting a gene already known to be strongly involved in various neurodegenerative diseases [33]. The effect of this variant on the age of onset was also demonstrated in an independent FTLD cohort, negative for mutations in *GRN/C9orf72*.

## 2. Results

### 2.1. Subjects

Targeted genetic screening for the presence of variants in the coding regions of 98 candidate lysosomal genes was performed on a total of *n* = 127 subjects belonging to 101 genealogically unrelated pedigrees (*n* = 74 *GRN* mutation carriers including 16 pre-symptomatic subjects and *n* = 53 *C9orf72* pathological expansion carriers, including one pre-symptomatic subject) (Table 1).

### 2.2. Single Variants Association Analysis

First, we estimated, by simple linear regression analysis, the association between age of onset and genetic variants in lysosomal genes. To this aim, only a patient for each genealogically unrelated pedigree was considered for the single variant association study (*n* = 99 *GRN* and *C9orf72* patients). The demographic and clinical characteristics of patients included in the single variant association analysis are summarized in Table 2. The sample is composed of 49% of *C9orf72* and 51% of *GRN* mutation carriers. Age and age of onset were not significantly different in sex and *GRN* and *C9orf72* patients’ groups.

The linear regression analysis revealed two genetic variants with a nominal *p*-value < 0.05 (Appendix A). After FDR correction, only rs1043424 (*PINK1*, c.1562A>C, p.Asn521Thr) resulted significantly associated with the age of onset (Table 3). Considering the age of onset as a dichotomous trait (with early onset ≤ 65 years considered as a risk trait and coded as 1), none of the variants resulted associated with onset after FDR correction (Table 3 and Appendix A).

In our sample, the rs1043424 *PINK1* variant determines a decrease in the age of onset from an average of 63 years for homozygous wild-type (AA, N = 52) to a mean age of 57 for heterozygotes (CA, N = 43) and 49 for the homozygous risk allele (CC, N = 4) (Figure 1). The recessive model showed a significant association with age of onset comparing CC homozygous versus A-carriers (Beta-coefficient = −11.1, CI = −20.96 ÷ −1.2, *p* = 0.029). The rs1043424 genotypes are equally distributed according to *C9orf72/GRN* mutation (*p* = 0.79) and sex (*p* = 0.84) (data not shown).

### 2.3. Replication Analysis

The *PINK1* p.Asn521Thr variant was tested in an independent group of 195 DNA samples from Italian unrelated FTLD patients, negative for mutations in *GRN/C9orf72.* Genotypes were available to us from the FTD-GWAS data set [8]. The *PINK1* variant showed a significant association considering the recessive model with a decrease in the age of onset from an average of 63.3 years for A-carriers (AA + CA, N = 178) to a mean age of 57.8 for homozygous risk allele carriers (CC, N = 17) (*p*-value = 0.036, Beta coefficient = −5.5, CI 95% = −10.7 ÷ −0.35). The rs1043424 genotypes (AA + CA versus CC carriers) are equally distributed according to sex (*p* = 0.25) (data not shown).

### 2.4. Cox Proportional Hazard Model and Kaplan–Meier Analysis

Cox proportional regression analysis on the whole group of *GRN* and *C9orf72* mutation carriers (Table 1) revealed that, after FDR correction, only rs1043424 (*PINK1*, c.1562A>C, p.Asn521Thr) was significantly associated with the age of onset, with a corrected *p*-value = 0.032 (Table 4). This analysis suggested that, with respect to wt-allele, each allele of the variant could increase hazard by 2.11 times (95% CI: 1.40 ÷ 3.20).

Moreover, to obtain the median age of onset for the different *PINK1* rs1043424 genotypes, we used the Kaplan–Meier estimate (Figure 2): the median age of onset for the different genotypes is 61 years (95% CI: 60 ÷ 65) in homozygous wild type AA carriers, 59 years (95% CI: 57 ÷ 62) in heterozygous CA carriers and 49 years (95% CI: 48 ÷ NA *; * the number of observations is too small to obtain an estimate for the upper limit of the confidence interval) in homozygous CC carriers (p log rank-test < 0.0001). This effect was maintained considering *GRN* and *C9orf72* as distinct groups (*GRN*: p log rank-test = 0.015; *C9orf72:* p log rank-test = 0.00075) (Appendix A).

### 2.5. Variant Interpretation

The identified *PINK1* missense variant was further characterized by bioinformatic prediction tools (Table 5).

The *PINK1* p.Asn521Thr variant is classified as benign according to the American College of Medical Genetics (ACMG) criteria. This variant has a population frequency of 29%, according to gnomAD, and it is predicted as benign by the majority of the in silico prediction tools. It scored 14.23 (pathogenic if >20) with the CADD tool and was predicted as benign/tolerated by Polyphen-2, SIFT, FATHMM, and Mutation Taster. The position scored 0.32 with GERP, meaning that mutations in this particular site follow the neutral rate of evolution and that the site is not particularly conserved. According to I-Mutant, the variant should increase PINK1 protein stability (ΔΔG = 1.24), while according to MUpro, it should decrease the protein stability (ΔΔG = −0.816). The variant was predicted to be structurally neutral for the PINK1 protein by Missense3D-DB. In public databases, p.Asn521Thr was described on ClinVar as Benign/Likely Benign (2-star classification score) for the congenital disorder of glycosylation and Parkinson’s disease (PD) (autosomal recessive, early onset). In Human Gene Mutation Database (HGMD), this variant was described as a risk variant for type 2 diabetes and for early onset Alzheimer’s Disease (AD).

### 2.6. Gene-Based Association

We performed an explorative aggregation analysis to evaluate the cumulative effects of multiple rare variants in a gene. None of the genes was statistically significant. However, the explorative analysis highlighted two genes (*DPP7* and *RTN4*) enriched by rare variants in early-onset with respect to late-onset patients (Appendix A) and three genes (*DYNC1H1*, *ATP13A2, ATP6V0A1*) presenting rare variants more frequent in late-onset patients (Appendix A). In particular, five variants mapping *DPP7* are exclusively carried by early onset patients, whereas p.Ala431Val is shared between the age of onset groups (Appendix A). All six different variants identified in *RTN4* were missense (Appendix A): four were exclusively carried by early-onset patients, and two variants were exclusively carried by late-onset patients.

## 3. Discussion

In diseases presenting with high phenotypical variability, even in the presence of the same genetic alteration, searching for modulators/modifiers of this variability is suggested. In FTLD, one of the most heterogeneous clinical features is the age of onset, even in the same family pedigree [26]. Due to the high hereditability of this group of disorders [5], some genetic studies have been performed in this field, showing the presence of genes potentially affecting the age of onset [27,28,29,34,35]. However, this issue is still far from being fully addressed. Thus, the aim of our study was to look for genetic variants affecting the age of onset in FTLD patients carrying *GRN* or *C9orf72* mutations because of the existence of common pathogenetic mechanisms underlying these genetic pathologies, in particular, the lysosomal dysfunction. We then selected a panel of 98 genes involved in different ways in lysosomal function and highly expressed in brain regions of interest, and by means of NGS technology, we identified the variants present in our patients and performed association studies using the age of onset as the dependent variable. The *PINK1* variant p.Asn521Thr (rs1043424) was the only variant modulating the age of onset in FTLD patients carrying *GRN* mutations or *C9orf72* expansion.

PINK1 is a protein kinase that interacts with Parkin, an E3 ubiquitin ligase, to eliminate dysfunctional mitochondria by the autophagy/lysosomal pathway [33,36]. The *PINK1* gene was discovered by linkage analysis to cause early-onset PD with recessive transmission patterns [37]: two homozygous mutations were identified, both affecting the PINK1 kinase domain. Similarly, a homozygous mutation causing truncation of the C-terminus of the PINK1 protein outside the kinase catalytic domain was described to cause early onset PD [38]. *PINK1* mutations at heterozygous state have afterward been associated with susceptibility to PD [39,40]. Heterozygous variants in *PINK1* (including the *PINK1* rs1043424 variant) have also been detected in patients affected by early-onset AD and Lewy Body Disease [41,42,43], although the role of *PINK1* in the susceptibility to these disorders is yet to be clarified.

Herein, we provided evidence that genetic variation in the *PINK1* gene might modulate disease phenotype, being associated with an earlier age of onset. Specifically, in *GRN/C9orf72* FTLD, we observed that (i) the *PINK1* rs1043424 C allele is significantly associated with the age of onset; (ii) every risk C allele could increase hazard by 2.11%; (iii) the estimated median age of onset in homozygous risk allele carriers (CC) was 10 and 12 years earlier than heterozygous (AC) and wild type homozygous carriers (AA). Moreover, a replication study in *GRN/C9orf72* negative FTLD patients confirmed that the rs1043424 C allele was associated with earlier disease onset (−5.5 years in CC versus AA + AC).

These data suggest that in patients with a vulnerable biological background, variation in *PINK1* (even if not predicted to be pathogenic) might have an additive effect exacerbating disease phenotype. Under pathological conditions, such as oxidative stress, the altered membrane potential of mitochondria leads to PINK1 accumulation on mitochondria with Parkin recruitment and activation; Parkin then ubiquitinates mitochondrial proteins, a signal for mitophagy. Pathogenic mutations in *PINK1* or *Parkin* alter this interaction, resulting in dysfunctional mitochondria accumulation. Interestingly, a recent study demonstrated that the *PINK1* p.Asn521Thr variant affects mitophagy slightly, reducing mitophagy induction compared to WT *PINK1* [44]. Growing evidence supports the contribution of mitophagy impairment to several human neurodegenerative diseases such as PD, AD, and Amyotrophic lateral sclerosis (ALS) [33,45]. In fact, a hallmark of AD is the accumulation of dysfunctional mitochondria, and dysregulated levels of PARKIN and PINK1 were detected [46]. Mitochondrial dysfunction has also been associated with ALS, and altered expression levels of mRNA and protein for PINK1 have been identified in human ALS patient muscle [47].

Concerning FTLD, a decreased expression of a mitochondrial module in frontal cortex tissue from FTD-TDP patients was reported [48], and a very recent study has shown that in brains of FTD patients carrying different *GRN* mutations, mitochondrial dysregulation is detected in neurons at variance with *MAPT* mutations carriers, where other cellular processes are affected [49]. In particular, proteins of the oxidoreductase complex are down-regulated, especially those of respiratory chain complex 1. We can hypothesize that if this mitochondrial dysregulation leads to an altered membrane potential, the PINK1 function becomes particularly relevant to remove dysfunctional mitochondria, and a polymorphism such as p.Asn521Thr may affect in some way this function. In addition, progranulin deficiency, which alters the lysosomal function necessary for mitophagy, may also result in abnormal mitochondria accumulation. As for *C9orf72* expansions, disruption of endoplasmic reticulum-mitochondria tethering and signaling has been reported in *C9orf72* FTD patients [50], underlying also in this genetic context mitochondrial pathway alteration.

One of the limitations of our study is the relatively small number of mutation carriers included. Moreover, we focused our investigation on patients coming from a specific geographical area, i.e., Northern Italy, where a founder effect was reported for a specific *GRN* mutation [51]. This is a pilot study, and further validation in larger groups of these rare forms of genetic neurodegenerative disease is needed.

Understanding the potential mechanisms behind the observed modulating effect of the *PINK1* gene in FTLD might prove critical for identifying biomarkers and/or designing drugs to modify the age of onset, especially in *GRN/C9orf72*-driven disease. Finally, the detected *PINK1* rs1043424 could be used to better predict the age of onset in asymptomatic carriers in preventive clinical trials and for genetic counseling.

## 4. Materials and Methods

### 4.1. Participants

This retrospective study was carried out on DNA from a total of *n* = 127 subjects (*n* = 74 *GRN* mutation carriers, *GRN*, and *n* = 53 *C9orf72* pathological expansion carriers, *C9orf72*). Patients were enrolled at the MAC Memory Clinic IRCCS Fatebenefratelli, Brescia, and at the Neurology 5/Neuropathology Unit, IRCCS Besta, Milan. Clinical diagnosis of FTLD was made according to international guidelines [52,53]. Diagnosis of NCL was made by ultrastructural examination of skin biopsy [22]. Participants signed informed consent for blood collection and biobanking, as approved by the local ethics committee (approval number 2/1992; 26/2014). *C9orf72* and *GRN* genetic screening was performed previously as described in [17,19,54]. Demographic and clinical characteristics are reported in Table 1. The study protocol was approved by the local ethics committee (Prot. N. 44/2018).

The replication study was performed in an independent group of 195 DNA samples from unrelated FTLD patients, negative for mutations in *GRN/C9orf72.* Genotypes were available to us from the FTD-GWAS data set [8]. Specifically, we had access to PINK1 rs1043424 genotypes of a total of 195 samples from IRCCS Fatebenefratelli, Brescia, and Fondazione IRCCS Istituto Neurologico “Carlo Besta”, Milano (Italy). Mean (±standard deviation) age of onset was 62.8 (±10.3) years with a male-to-female ratio of 89/106. The study protocol was approved by the local ethics committees “Comitato Etico IRCCS San Giovanni di Dio Fatebenefratelli” (Prot. N. 28/2013, Prot. N. 8/2017) and “Comitato Etico della Fondazione IRCCS Istituto Neurologico Carlo Besta” (Prot. N. 31-09/12/2009).

### 4.2. Gene Selection

A total of 37 search terms linked to lysosomal functions were evaluated and chosen among all the available ontology and pathways terms in the GSEA portal [https://www.gsea–msigdb.org/gsea/index.jsp (accessed on 20 November 2019)]. From these 37 search terms, a list of 557 unique genes was created. All the genes were then filtered depending on their expression levels in multiple brain regions (i.e., amygdala, caudate basal ganglia, cortex, frontal cortex, hippocampus, and putamen basal ganglia). Gene expression levels were downloaded from the GTEx portal [https://gtexportal.org/home/ (accessed on 20 November 2019)]. Expression percentiles were then calculated, and all the genes that fell above the 90th percentile for all the brain regions of interest were selected. A list of 228 genes was obtained. The final genes set was created by selecting the top 98 genes from the list based on their overall expression levels (Appendix A).

### 4.3. Genetic Analyses

The entire coding regions of the 98 candidate genes were analyzed by amplicon-based target enrichment and Next-Generation Sequencing (NGS) of the exons and exon-intron boundaries on an Illumina^®^ MiSeq platform (Illumina, San Diego, CA, USA). The quality assessment of gDNA was performed on a 0.8% agarose gel, and gDNA was quantified with a Qubit dsDNA HS Assay Kit (Thermo Fisher Scientific, Waltham, MA, USA). A total of 200 ng of gDNA was used for library preparation with Illumina DNA Prep with Enrichment kit (Illumina, Inc., San Diego, CA, USA). gDNA was tagmented, amplified, and purified with AMPure XP Beads (Beckman Coulter, Inc., Brea, CA, USA). The size, quality, and quantity of libraries were assessed with a High Sensitivity DNA kit on a Bioanalyzer instrument (Agilent Technologies, Santa Clara, CA, USA). A 12 pM sample of the pooled library was loaded on a MiSeq reagent cartridge v3 and sequenced on an Illumina MiSeq platform.

### 4.4. Bioinformatic Analysis: Data Pre-Processing, Mapping, and Variant Calling

The overall data quality was evaluated with FastQC (version 0.11.9) [http://www.bioinformatics.babraham.ac.uk/projects/fastqc/, accessed on 11 August 2022]. The raw data were cleaned using Trimmomatic (version 0.38) [55] by removing adapters and reads of poor quality. The high-quality reads were then aligned versus the reference genome (hg19) using bwa software (mem algorithm, 0.7.17-r1188) [56]. A coverage analysis was performed with the Depth Of Coverage module of Genome Analysis Toolkit (GATK, version 3.8.1) software [https://gatk.broadinstitute.org/, accessed on 11 August 2022]. All samples showed coverage of at least 50X on all 98 panel genes. Subsequently, duplicated read marking was performed using Picard [57], and the single-nucleotide variant (SNV) and insertion/deletion (INDEL) calling were carried out using the Haplotype Caller module of GATK (version 4.1.8) software over the target region. The Single-Nucleotide Polymorphism Database (dbSNP; v151) was used as the variant reference database. A genomic variant call format (gVCF; version 4.1.8) has been created for each sample. All the gVCF files were then merged in a single multisample gVCF. Genetic variants were filtered according to GATK hard-filtering guidelines. Variants with total reads count ≤ 20, alternative allele depth ≤ 10, allele balance ˂ 25%, and call rate < 80% have been excluded. The location and effect of each variant on gene function were predicted using SnpEff [58] and ANNOVAR (version 2020-06-08) [59] programs. For statistically significant variants, additional in silico prediction tools were employed in order to better understand their effect, as suggested in the American College of Medical Genetics (ACMG) sequence variants interpretation guidelines [60]. The chosen tools were CADD (v1.6) [https://cadd.gs.washington.edu/ (accessed on 11 August 2022)] [61], Polyphen-2 (version 2.2.3) [http://genetics.bwh.harvard.edu/pph2/index.shtml (accessed on 11 August 2022)] [62], Sift [https://sift.bii.a–star.edu.sg/ (accessed on 11 August 2022)] [63], FATHMM (v2.3) [http://fathmm.biocompute.org.uk/ (accessed on 11 August 2022)], Mutation Taster [https://www.mutationtaster.org/ (accessed on 11 August 2022)] and GERP [http://mendel.stanford.edu/SidowLab/downloads/gerp/ (accessed on 11 August 2022)] [64]. For protein stability predictions, I-Mutant (v2.0) [https://folding.biofold.org/i–mutant/i–mutant2.0.html (accessed on 11 August 2022)] [65], MUpro (v1.0) [http://mupro.proteomics.ics.uci.edu/ (accessed on 11 August 2022)] [66] and Missense3D-DB (v.1.5.1) [http://missense3d.bc.ic.ac.uk:8080/home (accessed on 11 August 2022)] [67,68] were used. Variants population frequencies were downloaded from gnomAD (2.1.1) [https://gnomad.broadinstitute.org/ (accessed on 11 August 2022)] database, specifically from the NFE (Non-Finnish Europeans) subset. Additional annotations were obtained from ClinVar [https://www.ncbi.nlm.nih.gov/clinvar/ (accessed on 11 August 2022)], OMIM [https://www.omim.org/ (accessed on 11 August 2022)] and HGMD [https://www.hgmd.cf.ac.uk/ac/index.php (accessed on 11 August 2022)]. Proteins 3D models were created with PyMOL (v2.5) [https://pymol.org/2/ (accessed on 11 August 2022)].

### 4.5. Statistical Analysis

Continuous variables (age and age of onset) are presented as mean and standard deviation and categorical variables as numbers and percentages. The normality of continuous features was checked with Kolmogorov–Smirnov test. Between-group comparison of continuous variables was performed using one-way ANOVA or *t*-test. Categorical data were compared between groups with the Chi-squared test.

To identify genetic variants of which the allele frequencies vary systematically as a function of the age of onset values as a phenotypic trait, we performed a single variants association analysis. Specifically, we considered only non-synonymous variants (missense, splicing, stop-gain, stop-loss, conservative or frameshift ins/del variants) with low frequency (0.01 < MAF < 0.05) or common (MAF > 0.05) in our dataset. Moreover, we used linear regression analysis considering the quantitative trend of the onset. We also performed a logistic regression analysis for a discrete trait, grouping patients into two classes according to the age of onset, early (≤65 years, coded as 1) and late (>65 years, coded as 0). Subjects’ sex and genetic group (*C9orf72* or *GRN*) were included as covariates in the models, and *p*-values were corrected using FDR [69]. The analyses were performed using PLINK (version v1.90b6.21) [70].

To assess if each allele of the selected variants affects the age of onset, we used a Cox proportional hazard regression model [71]. This model was implemented, including all 127 subjects (patients and pre-symptomatic subjects) (Table 1), censoring the pre-symptomatic to their age of sampling. Subjects’ sex and genetic group (*C9orf72* or *GRN*) were included as covariates in the model. Moreover, to adjust for relatedness, we created an indicator number for each family, and we used the coxme function of the R “coxme” package [72] [https://cran.r–project.org/web/packages/coxme/index.html (accessed on 30 March 2022)].

The different incidence of the disease among genotypes of significant variants was shown using Kaplan–Meier curves [73]. Kaplan–Meier curves were implemented including all 127 subjects (patients and pre-symptomatic subjects) (Table 1), censoring the pre-symptomatic to their age of sampling. The analyses were carried out using R software (“survival”, “survminer”, “Rccp” packages) [https://cran.r–project.org/web/packages/survival/index.html (accessed on 24 February 2022)] [https://cran.r-projct.org/web/packages/survminer/index.html (accessed on 24 February 2022); https://cran.r-project.org/web/packages/Rcpp/index.html (accessed on 24 February 2022)].

To evaluate the cumulative effects of multiple rare variants in a gene, gene burden analysis was performed. Only missense, splicing, stop-gain, stop-loss, conservative, or frameshift ins/del variants (the so-called “non-synonymous” variants) with MAF < 0.01 were collapsed into a single gene. Genes mapped by only one rare non-synonymous variant were not considered. To test whether there is a higher excess of rare non-synonymous variants in early-onset in comparison with late-onset patients, we applied both the burden (b.burden) and the SKAT-O test [http://genome.sph.umich.edu/wiki/EPACTS, accessed on 11 August 2022] as implemented in EPACTS software [74].

## Figures and Tables

**Figure 1 ijms-23-12847-f001:**
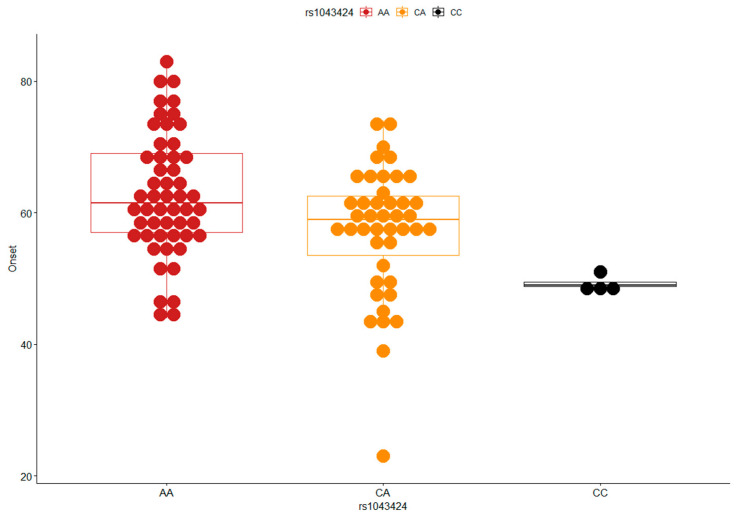
Boxplot of the age of onset distribution according to rs1043424 *PINK1* genotypes. Homozygous wild-type (AA), heterozygotes (CA), and homozygous risk allele (CC).

**Figure 2 ijms-23-12847-f002:**
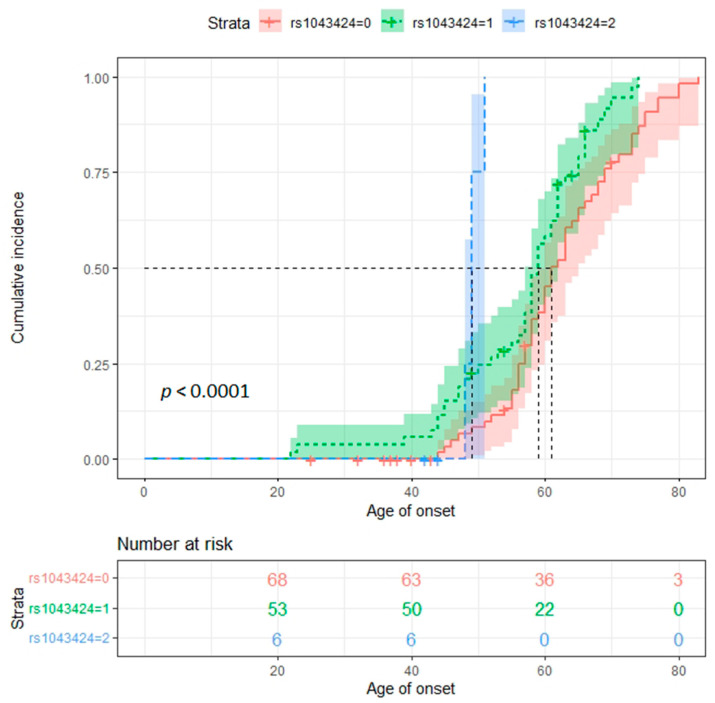
Kaplan–Meier curve showing disease incidence in the three rs1043424 *PINK1* genotypes.

**Table 1 ijms-23-12847-t001:** Clinical and demographic characteristics of subjects included in the study.

		**PATIENTS**		**UNAFFECTED**	**All**	***p*-Value** **Comparisons**
N.		110		17		
N.		*GRN*	*C9orf72*	*GRN* + *C9orf72*		
		58 (45.7%)	52 (40.9%)	17 (13.4%)	127	
Age		61.0 (11.4)	60.5 (9.5)	47.8 (13)	59 (11.7)	Overall: 6.72 × 10^−5 a^ *C9orf72* vs. U: 1.70 × 10^−4 a^ *GRN* vs. U: 7.60 × 10^−5 a^ *GRN* vs. C90rf72: 1 ^a^
Sex	M	35 (60%)	25 (48%)	10 (58.8%)	70	0.41 ^b^
	F	23 (40%)	27 (52%)	7 (41.2%)	57	
Ag of Onset		58.9 (11.1)	59.6 (9.5)	/	59.2 (10.4)	0.72 ^a^

Mean (±Standard Deviation) was computed for continuous variables, while N (%) was computed for categorical variables; ^a^ One-way ANOVA test with posthoc Bonferroni correction; ^b^ chi-squared test; PATIENTS, patients with frontotemporal lobar degeneration due to *GRN* or *C9orf72* mutations; UNAFFECTED, pre-symptomatic patients; U, Unaffected.

**Table 2 ijms-23-12847-t002:** Clinical and demographic characteristics of patients included in the single variant association analysis.

		*GRN* Patients	*C9orf72* Patients	All	*p*-Value Comparisons
N.		51	48	99	
Sex	M	29 (57%)	23 (48%)	52	0.37 ^b^
	F	22 (43%)	25 (52%)	47	
Age of Onset		60.1 (10.3)	56.6 (9.5)	59.9 (9.9)	0.82 ^a^
Early onset (age of onset ≤ 65 years)		35 (69%)	34 (71%)	69 (70%)	0.81 ^b^

Mean (±Standard Deviation) was computed for continuous variables, while N (%) was computed for categorical variables; ^a^
*t*-test; ^b^ chi-squared test; *GRN* Patients, patients carrying *GRN* null mutations; *C9orf72* Patients, patients carrying *C9orf72* pathological expansion.

**Table 3 ijms-23-12847-t003:** Low frequency and common genetic variants associated with age of onset.

SNP	GENE	Location	c.pos	p.pos	P Linear	P Linear FDR	Beta Linear	CI (95%) Linear	P Logistic	P Logistic FDR	OR Logistic	CI (95%) Logistic
rs1043424	*PINK1*	missense	c.1562A>C	p.Asn521Thr	5.28 × 10^−4^	0.044	−5.91	−9.13 ÷ −2.68	0.042	1	2.38	1.03 ÷ 5.48

C.pos, coding position; p.pos, protein position; P, Beta, and CI Linear, *p*-value, beta coefficient, and confidence interval derived from linear regression; Beta coefficient is referred to minor allele C; P, OR, and CI Logistic, *p*-value, odds ratio, and confidence interval derived from logistic regression.

**Table 4 ijms-23-12847-t004:** The Cox proportional hazard regression results for the association of missense variants with age of onset.

SNP	GENE	Location	c.pos	p.pos	*p* Value	*p* Value FDR	HR	95% CI
rs1043424	*PINK1*	missense	c.1562A>C	p.Asn521Thr	4.00 × 10^−4^	0.032	2.11	1.40 ÷ 3.20

C.pos, coding position; p.pos, protein position; *p* Value and HR, *p*-value and hazard ratio derived from Cox proportional hazard regression.

**Table 5 ijms-23-12847-t005:** *PINK1* missense variant.

GENE	VARIANT	gnomAD_NFE (Genome/Exome)	CADD	Poly- phen2	GERP	FATHMM	Mutation Taster	ΔΔG MUPro and I-Mutant	Missense 3D-DB
*PINK1*	p.Asn521Thr	0.281/0.278	14.23	B	0.32	B	Polymorphism	−0.816/1.24	S. Benign

gnomAD_NFE, genome aggregation database non-Finnish European; CADD, combined annotation dependent depletion; Poly-Phen2, polymorphism phenotyping v2; B, benign; GERP, genomic evolutionary rate profiling; FATHMM, functional analysis through hidden markov models; ΔΔG, protein stability free-energy change; S, structurally.

## Data Availability

The raw data supporting the conclusions of this article are openly available in the Zenodo Data Repository at doi:10.5281/zenodo.7040532 [75].

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
