# Peer review of "The PINK1 p.Asn521Thr Variant Is Associated with Earlier Disease Onset in GRN/C9orf72 Frontotemporal Lobar Degeneration"

_ijms, 2022, doi:10.3390/ijms232112847_

Round 1

Reviewer 1 Report

  • Major concerns to be addressed:  

  • - The use of the term “anticipates” is not at all appropriate for this manuscript. In relation to genetics, anticipation describes an increasingly severe disease phenotype from one generation to the next, which is not described in this study. Suggest change to “is associated with” in the title and throughout the manuscript. 

  • - The extremely low number of C/C carriers in the cohort makes it very difficult to assess the accuracy/strength of the reported association. It is understood that this is a pilot study, however finding an association with age of onset in a replication cohort is needed to be sure of the authors' claims. Failing that, functional assay(s) that provide evidence of a modulating effect of the SNP in GRN or C9orf72 carriers is needed. Are fibroblasts or iPSCs available to look at effects of the SNP on mitophagy in the AA versus AC or CC carriers? Or can the authors use cell models with overexpressed PINK1 variants to test for effects on mitophagy or lysosomes? 

  • - There is no discussion in the introduction of the possibility of FTLD being a polygenic disease.

  • - Were all SNPs in HWE? 

  • - Section 2.2 – why was only one patient per pedigree chosen for association between age of onset and genetic variants? 

  • - Line 175 – age of onset is the dependent rather than the independent variable. 

  •  

Minor suggested edits and changes for an English-speaking audience: 

  • - Abstract – should 10/12 years earlier be 10-12? 

  • - References to genes (e.g., GRN, C9orf72, PINK1) should be italicised. 

  • - Introduction – suggest change “On a clinical prospect,” to “Clinically,” 

  • - Line 45 carrier to carriers 

  • - Line 57 – remove “and the genes coding for” (unnecessary) and “for” from and for tau (MAPT) 

  • - Line 55 – add “and” before iii) 

  • - Line 61 ; in “the” central nervous system it modulates … 

  • - Line 63 – evidences to evidence 

  • - Line 64 – and also affects 

  • - Line 68 – share both molecular mechanisms, ….. [REFS], and  

  • - Line 76 Herein, we 

  • - Line 76 remove both 

  • - Line 77 and to or 

  • - Line 78 remove “a” before targeted  

  • - Asn521Thr should be referred to as p.Asn521Thr throughout 

  • - In Table 2, Male GRN carriers should be (57%) 

  • - Line 104, using age of onset as a dichotomous trait – what are the 0 = X and 1 = X? over and under age 65?  

  • - Table 3 – what are the confidence intervals for your OR? 

  • - Line 156 – early onset with respect to  

  • - Line 164 – mandatory is not appropriate. Suggested, useful, informative. 

  • - Line 169 being to fully 

  • - Line 187 – add space between Lewy Body Disease and refs 

  • - Line 188 “has still” to “is yet” 

  • - Line 209 – “ad to and 

  • - Line 207/208 – what is meant by this condition, FTLD? Please specify. 

  • - Line 211 – “necessary to mitophagy” to necessary for mitophagy”  

  • - Line 214 – “pathways” to “pathway” 

  • - Line 220 – “of PINK1 gene” to “of the PINK1 gene” 

  • - Line 242 – “al” to “all” 

  • - Line 244 – “Genes” to “Gene” 

  • - Line 268 - “al” to “all” 

  • - Line 269 – “has been” to “was” 

  • - Line 270 – “have been” to “were” 

  • - Line 275 “hard filteing” to “hard filtering” 

  •  

Reviewer 2 Report

I think the manuscript can be accepted as it is.

Reviewer 3 Report

Rossi et al undertake a retrospective analyses to determine what additional snps are associated with age of onset in FTLD. The authors found that a snp in PINK1 was positively associated with early onset in patients with GRN/C9orf72-associated disease. Overall this was an interesting study to target the lysosome pathway as degradation pathways are critical to the proper functioning of brain cells. The study is limited, as the authors pointed out, by the sample size and geographical diversity in included patients. Nevertheless, given the association of progranulin in lysosomal functions, it makes some sense that PINK1 can be implicated. Overall the manuscript is clear and well put together; minor spelling and grammar edits are required (e.g. line 242, "Al" should be "All"). Additionally, more exposition on the potential role of PINK1 in lysosomal function and mitophagy as they relate to neurodegeneration would be welcomed. Are there any studies looking at altered mitochondrial function in FTLD?

Round 2

Reviewer 1 Report

The authors have addressed all of my concerns, of particular importance the inclusion of a replication study that confirms the association identified between this SNP with an earlier age at onset in FTD/FTLD.